# Evaluating Human Photoreceptoral Inputs from Night-Time Lights Using RGB Imaging Photometry

**DOI:** 10.3390/jimaging5040049

**Published:** 2019-04-16

**Authors:** Alejandro Sánchez de Miguel, Salvador Bará, Martin Aubé, Nicolás Cardiel, Carlos E. Tapia, Jaime Zamorano, Kevin J. Gaston

**Affiliations:** 1Environment and Sustainability Institute, University of Exeter, Penryn, Cornwall TR10 9FE, UK; 2Departamento de Física de la Tierra y Astrofísica, Instituto de Física de Partículas y del Cosmos (IPARCOS), Universidad Complutense, 28040 Madrid, Spain; 3Instituto de Astrofísica de Andalucía, Glorieta de la Astronomía, s/n, C.P.18008 Granada, Spain; 4Departamento de Física Aplicada, Universidade de Santiago de Compostela, 15782 Santiago de Compostela, Galicia, Spain; 5Physics Department, CEGEP de Sherbrooke, Sherbrooke, QC J1E 4K1, Canada

**Keywords:** light pollution, imaging, artificial light at night, night-time lights, DSLR cameras, RGB sensors, non-visual effects of light, circadian phototransduction

## Abstract

Night-time lights interact with human physiology through different pathways starting at the retinal layers of the eye; from the signals provided by the rods; the S-, L- and M-cones; and the intrinsically photosensitive retinal ganglion cells (ipRGC). These individual photic channels combine in complex ways to modulate important physiological processes, among them the daily entrainment of the neural master oscillator that regulates circadian rhythms. Evaluating the relative excitation of each type of photoreceptor generally requires full knowledge of the spectral power distribution of the incoming light, information that is not easily available in many practical applications. One such instance is wide area sensing of public outdoor lighting; present-day radiometers onboard Earth-orbiting platforms with sufficient nighttime sensitivity are generally panchromatic and lack the required spectral discrimination capacity. In this paper, we show that RGB imagery acquired with off-the-shelf digital single-lens reflex cameras (DSLR) can be a useful tool to evaluate, with reasonable accuracy and high angular resolution, the photoreceptoral inputs associated with a wide range of lamp technologies. The method is based on linear regressions of these inputs against optimum combinations of the associated R, G, and B signals, built for a large set of artificial light sources by means of synthetic photometry. Given the widespread use of RGB imaging devices, this approach is expected to facilitate the monitoring of the physiological effects of light pollution, from ground and space alike, using standard imaging technology.

## 1. Introduction

Interest in the physiological effects of artificial light at night has steadily grown in recent years. The alteration of natural cycles of light and darkness, brought about by the widespread use of artificial light, has been shown to contribute to circadian rhythm disruption, sleep disorders, and other potentially relevant public health outcomes [1,2,3,4,5,6,7,8,9]. Increasing awareness of the unwanted effects of light pollution and the quest for a sustainable approach to outdoor lighting have fostered the search for practicable methods of evaluating human exposure to artificial light at night.

Although the existence of non-image-forming effects of light on the human body had long been known, it was not until the turn of the century that a major breakthrough in understanding these interactions was achieved, after a novel non-rod, non-cone type of photoreceptor was first predicted and then found in the mammal retina [10,11,12,13]. This recently discovered photoreceptor, an intrinsically photosensitive subset of different varieties of retinal ganglion cells (ipRGC), expresses a light-detecting photopigment, melanopsin, not found in classical rods and cones. The ipRGC project via the retinohypothalamic tract to the suprachiasmatic nuclei of the hypothalamus, where the master oscillator of the central circadian clock is located. With a smaller absolute responsivity to light than rods and cones and a spectral sensitivity band centered about 484 nm, the ipRGC are, as presently best understood, the main photoreceptors for the daily entrainment of the circadian clock. 

The ipRGC output, however, is not entirely determined by melanopsin light detection. The ipRGC synapses receive additional inputs from both rods and cones, through the intermediate layers of horizontal, bipolar, and amacrine retinal cells [14,15]. The overall ipRGC response is then a generally nonlinear combination of the excitations of the five classes of photoreceptors (rods; S-, L- and M-cones; and the ipRGC themselves), whose particular details depend on the physiological outcome under study. In order to make possible the quantitative comparison of experimental results and the realization of meta-analyses of clinical and epidemiological studies, it has been strongly recommended that the precise spectral power density of the irradiance incident on the eye, as well as the detailed exposure conditions for each experimental session, should be carefully recorded. In the absence of such detailed spectral information, at least some quantities reflecting the excitations of the five individual classes of photoreceptors should be calculated and reported [16]. The suggested quantities of choice are the in-band photoreceptor irradiances (W m^−2^) at the subject’s cornea, i.e., the incident spectral irradiance (W·m^−2^·nm^−1^) weighted by the spectral sensitivity function of the corresponding photoreceptor (dimensionless) and integrated across wavelengths (nm) [16,17].

The accelerated pace of substitution of the traditional gas-discharge lamps (mostly high-pressure sodium vapor and metal halides) by LED streetlights, with the associated shifts in illumination levels and spectral power density distributions, has reinforced the need for monitoring the changes in the nighttime light environment since some of their side-effects could be potentially relevant to human health. Wide-area sensing of public outdoor lighting for different light pollution research applications is routinely carried out using the radiometers already onboard Earth-orbiting satellites (see, e.g., [18,19,20,21,22,23,24,25]). However, most operating spaceborne radiometers with sufficient nighttime sensitivity are panchromatic [26,27,28,29,30], lacking the required spectral discrimination for assessing health effects. One way to partially overcome this limitation is the use of RGB imagery, such as the well-known images of the Earth at night obtained by astronauts on the International Space Station (ISS) with off-the-shelf digital single-lens reflex cameras (DSLR) from the Crew Earth Observations facility (CEO) [31]. After suitable camera calibration, the DSLR raw images allow the radiance reaching the ISS from the streetlights located within the camera field-of-view to be determined, with a spatial resolution of the order of tens of meters, and a limited but extremely useful RGB multi-spectral capability [32,33] (see also [34]). The use of DSLR cameras for light pollution research has expanded significantly in recent years, and its effectiveness has been demonstrated in applications such as all-sky night sky brightness monitoring [35,36,37,38,39], cloud reflection and screening studies [40], and nighttime lights dynamics assessment [41,42,43], among others. 

Remote sensing of nighttime lights is increasingly being used in public health research [44,45,46,47,48]. In particular, ISS imagery in the RGB bands has been instrumental for recent epidemiological studies regarding the association between light at night and some pathologies [49,50]. Whereas in [49] the R, G, and B signals were used as independent input variables, in [50] these signals were used to estimate several spectral indices deemed relevant to human health. A practical method for estimating band-weighted radiometric quantities using linear relationships to the B/G or G/R inputs has been described in [51]. In the present paper we significantly expand the scope of that method by (i) taking advantage of the additional non-redundant information available when using as input variables optimum linear combinations of the R, G, and B chromatic channels and (ii) applying this approach to determine, from RGB imagery, the values of the photoreceptor-weighted exposures in the photometric bands of sensitivity of the human rods; S-, L- and M-cones; and ipRGC. That way, RGB images can be used to assess the contribution of each light source present in the field of view to the relative excitation of the five human photoreceptors, which is presently believed to be the basic information required to evaluate the non-image-forming effects of light at night.

## 2. Materials and Methods

### 2.1. Photoreceptoral and RGB Spectrally-Weighted Radiant Quantities

Several radiant magnitudes are of interest for light pollution studies dealing with the physiological effects of optical radiation. The most basic one is the spectral radiance distribution at the entrance pupil of the eye (units W·m^−2^·sr^−1^·nm^−1^), which determines the spectral irradiance at each particular retinal location (W·m^−2^·nm^−1^) and hence the radiant spectral flux incident on individual photoreceptors (W·nm^−1^). The effective dose absorbed by each photoreceptor depends on the spectral composition of the incident beam weighted by the photoreceptor-specific spectral sensitivity. Human photoreceptors act like filters whose spectral sensitivity functions are determined by the absorption characteristics of the receptor opsins and the spectral transmittance of the pre-receptoral ocular media, including the retinal pigments. A set of standard spectral sensitivity functions for the five photoreceptors has been experimentally determined and proposed for general use [16,17]. Figure 1 displays the functions Cy(λ), Me(λ), Rh(λ), Ch(λ), and Er(λ), corresponding to the cyanopic (S-cone), melanopic (ipRGC), rhodopic (rod), chloropic (M-cone), and erythropic (L-cone) transmittance-corrected opsins, respectively. These "alfa-opic" functions are normalized to 1 at their peak, compliant with the SI criterion stated in CIE TN 003:2015 [17]. Thus, if L(λ) is the spectral radiance (W·m^−2^·sr^−1^·nm^−1^) incident on a given photoreceptor with spectral sensitivity Y(λ), the corresponding band-weighted radiance Y (in units " W·m^−2^·sr^−1^ within the Y(λ) band") will be:(1)$$Y=\int^{\;}Y\left(\lambda \right)L\left(\lambda \right)d\lambda ,$$ where Y(λ) stands for any of the alfa-opic Cy(λ), Me(λ), Rh(λ), Ch(λ), and Er(λ) functions, and Y is the effective radiance exciting the corresponding photoreceptor (which we will denote by Cy, Me, Rh, Ch, and Er, respectively). Note that the same type of weighted integral can be applied to other radiant magnitudes of interest, like irradiance, radiant flux, or radiant exposure, among others.

In an analogous way, the camera pixel readings in the R, G, and B bands are proportional to
(2)$$R=\int^{\;}R\left(\lambda \right)L\left(\lambda \right)d\lambda ,\;G=\int^{\;}G\left(\lambda \right)L\left(\lambda \right)d\lambda ,\;B=\int^{\;}B\left(\lambda \right)L\left(\lambda \right)d\lambda ,$$ where R(λ), G(λ), and B(λ) are the corresponding spectral sensitivity functions of the camera pixels spatially arranged in the Bayer matrix (see an example, corresponding to a Nikon D3s camera (Nikon, Sendai, Japan), in Figure 2). Note that after suitable calibration the camera pixel readings can be converted to band-weighted pixel radiant exposure (units J). The corresponding radiant flux (units W) can be obtained as the radiant exposure divided by the exposure time. This flux can, in turn, be used to compute the pixel irradiance (W·m^−2^) by dividing it by the pixel area. The radiance (W·m^−2^·sr^−1^) incident on each pixel can be computed by dividing the irradiance by the solid angle (sr) of the beam, which in turn depends on the camera lens numerical aperture (*f*-number). The band-weighted radiance incident on the camera is obtained by dividing the radiance incident on the pixel by the transmittance of the camera lens in the corresponding photometric band. These last two steps can be performed at once if there is information available about the band-weighted T-stops. The weighted irradiance at the input pupil of the camera can be deduced easily using standard photometric transformations.

The problem to solve can then be stated as follows: estimating the values of Y (Cy, Me, Rh, Ch, and Er) for the light sources within the field-of-view, from the camera readings R, G, and B.

### 2.2. Linear Estimation of the Photoreceptoral Weighted Radiances from RGB Signals 

An exact, analytic solution to the problem of determining the Y radiances from the R, G, and B signals would be obtained by finding the (Y-dependent) coefficients $c_{R}$, $c_{G}$, and $c_{B}$ that fulfill the equality
(3)$$Y=c_{R}R+c_{G}G+c_{B}B$$ for any arbitrary incident radiance L(λ). However, the general validity of Equation (3) requires that the corresponding weighting functions strictly conform to
(4)$$Y\left(\lambda \right)=c_{R}R\left(\lambda \right)+c_{G}G\left(\lambda \right)+c_{B}B\left(\lambda \right)$$ for all wavelengths λ. It is easy to see that there are no exact solutions for $c_{R}$, $c_{G}$, and $c_{B}$ allowing conformation to Equation (4) for any of the Cy(λ), Me(λ), Rh(λ), Ch(λ), and Er(λ) photoreceptor functions and the R(λ), G(λ), and B(λ) bands. In other words, the photoreceptoral Y(λ), considered as Hilbert space vectors, do not belong to the subspace spanned by {R(λ),G(λ),B(λ)}. Real values for $c_{R}$, $c_{G}$, and $c_{B}$ allowing approximate conformity to Equation (4) can be obtained by a linear least-squares fit (LSQ) of R(λ), G(λ), and B(λ) to Y(λ) over all wavelengths. However, for light pollution studies, this homogeneous LSQ fit over wavelengths, with all spectral λ equally weighted, turns out to be a suboptimal choice. Present-day outdoor lighting sources are manufactured using a restricted set of technologies and, in consequence, their light spectra belong to a finite set of basic patterns. This makes it reasonable to find the best fit of R(λ), G(λ), B(λ) to Y(λ) for these lamp spectra, such that the squared differences between the two sides of Equation (4) is minimized over the relevant lamp dataset.

To proceed in this way let us use a wide set of *N* different light lamps, with known spectral radiance distributions, $L_{i}\left(\lambda \right)$, *i* = 1,…,*N*, and let us determine by numerical integration the corresponding values of the band-weighted radiances $Cy_{i}$, $Me_{i}$, $Rh_{i}$, $Ch_{i}$, $Er_{i}$, and $R_{i}$, $G_{i}$, $B_{i}$, using Equations (1) and (2), a procedure commonly known in astrophysics as synthetic photometry. Ordering these band-weighted radiances as N×1 column vectors Cy, Me, Rh, Ch, Er, and R, G, B, respectively, we can rewrite Equation (3) in vector form, for the generic photoreceptor Y, as
(5)$$Y=c_{R}R+c_{G}G+c_{B}B$$ which is an overdetermined linear system of N equations with three unknowns ($c_{R}$, $c_{G}$, and $c_{B}$). This system has no exact solution, both for the fundamental reason outlined above regarding Equation (4) and because in practice all relevant radiant magnitudes involved are affected by measurement noise (in our case, the noise present in the experimentally measured lamp spectra used to compute the weighted radiances by means of synthetic photometry). However, Equation (5) can be solved in an LSQ sense by minimizing the squared differences between both sides of the equation over the whole lamp database. To do so, let us define the N×3 system matrix M=[R G B] formed by the three N×1 column vectors, and $c={\left(c_{R},c_{G},c_{B}\right)}^{t}$ the coefficient 3×1 column vector, where *t* stands for ’transpose’. In terms of these newly defined objects, Equation (5) can be rewritten as
(6)Y=M c that can be solved, in the LSQ sense, as
(7)$$\hat{c}=M^{+}\;Y$$ where $M^{+}={\left(M^{t}\;M\right)}^{-1}M^{t}$ is the Moore–Penrose pseudoinverse (size 3×N), and $\hat{c}$ denotes that the coefficients obtained by this procedure are a least-squares estimation, rather than an exact solution. Besides the option of calculating $M^{+}$ using the explicit formula indicated above, most commonly available mathematical software packages provide efficient built-in routines for computing the pseudoinverse of M, generally based on singular value decomposition (SVD) algorithms.

In practice, it is often convenient to slightly reformulate this problem in terms of dimensionless quantities. This situation arises when one wishes to compute the $\left(c_{R},c_{G},c_{B}\right)$ coefficients from lamp spectral data expressed in homogenous but otherwise arbitrary units. To that end, it is advantageous to normalize all weighted integrals to one of them, e.g., G, such that Equation (3) becomes
(8)$$Y/G=c_{R}R/G+c_{B}B/G+c_{G}$$

This expression lends itself well to graphical display since the dimensionless quantities (R/G,B/G, Y/G) can be interpreted as the (*x*, *y*, *z*) coordinates of a point representing a lamp in a three-dimensional (3D) space. All lamps in the database then form a cloud of points in this space. From this standpoint Equation (8) is the equation of a plane, and, for the whole set of lamps, finding the optimum $\left(c_{R},c_{G},c_{B}\right)$ coefficients is equivalent to the problem of finding the plane that best fits the cloud of points. The procedure to solve this issue is completely analogous to the one outlined above. Let us define the *G*-normalized vectors $Y_{G}=Y./G$, $R_{G}=R./G$, and $B_{G}=B./G$, where the symbol "./" denotes division element by element, such that Equation (8) in vector form can be rewritten as $Y_{G}=c_{R}R_{G}+c_{B}B_{G}+c_{G}$. Defining now the N×3 system matrix $M_{G}=\left[R_{G}\;1\;B_{G}\right]$, the set of Equations (6) becomes $Y_{G}=M_{G}\;c$, that can be solved in the LSQ sense as $\hat{c}=M_{G}^{+}\;Y_{G}$ by means of the pseudoinverse $M_{G}+={\left(M_{G}{}^{t}\;M_{G}\right)}^{-1}M_{G}{}^{t}$. The coefficients $\hat{c}$ thus determined are identical to the ones obtained using Equation (7).

In Section 3 we will take advantage of this *G*-normalized formulation to represent the results as two-dimensional scatter plots of Y/G versus the independent variable $X=c_{R}R/G+c_{B}B/G+c_{G}$. Note that, by definition, the straight line fit of Y/G versus X has unit slope and zero intercept. Note also that once the $\left(c_{R},c_{G},c_{B}\right)$ coefficients have been determined, the absolute, non-normalized value of Y for any lamp can be immediately obtained by multiplying its Y/G by its value of G.

### 2.3. Lamp Spectra Database

A total of 205 lamp spectra, comprising 28 compact fluorescent (CFL), 7 ceramic metal-halide (CMH), 3 T-type fluorescent (FL), 31 halogen (HAL), 6 high-pressure sodium vapor (HPS), 17 incandescent (INC), 97 light-emitting diode (LED) with CCT in the range 1900–7400 K, 15 metal halide (MH), and 1 mercury vapor (MV) lamps with native 5 nm wavelength resolution were used for this work. These belong to two different spectral libraries, the LSPDD database [52] and the LICA UCM spectra database [53]. Whereas the former consists of spectra measured in the laboratory, the latter relies mainly on field measurements of the spectra of outdoor lights. All spectra were linearly interpolated to 0.5 nm intervals before performing the required weighted integrations.

## 3. Results

The synthetic photometry calculations described by Equations (1) and (2) were performed for all the lamps contained in the sample used in this study. The results, displayed as point clouds in the *G*-normalized 3D space (R/G,B/G, Y/G) are shown in the left-hand column of Figure 3 for the cyanopic, melanopic, rhodopic, chloropic, and erythropic band-weighted radiances. These clouds of points can be fitted with reasonable efficacy by planes of the type described by Equation (8), determining the fit coefficients $\left(c_{R},c_{G},c_{B}\right)$ by the procedures above described. The results are displayed in the right-hand column of Figure 1, where the values of Y/G are plotted against the independent variable defined by optimum linear combination $X=c_{R}R/G+c_{B}B/G+c_{G}$. Table 1 summarizes the coefficient values and the standard deviation of the residuals of the fits.

## 4. Discussion

Our results suggest that the band-weighted radiances in the photoreceptoral bands Cy(λ), Me(λ), Rh(λ), Ch(λ), and Er(λ) can be estimated (in a least-squares sense) with reasonable precision from the R, G, and B weighted radiances. This opens the way for using calibrated off-the-shelf DSLR cameras to estimate, from the raw RGB images, the at-the-sensor cyanopic, melanopic, rhodopic, chloropic, and erythropic radiances. These are presently deemed to be the fundamental inputs required to describe the non-image-forming effects of light on human physiology. This work further develops the recently published [51], by taking advantage of the non-redundant information contained in the three RGB bands to find the optimal estimations of these physiologically relevant quantities.

The relative sizes of the $\left(c_{R},c_{G},c_{B}\right)$ coefficients in Table 1 are roughly indicative of the closeness of each photoreceptoral band to the R(λ), G(λ), and B(λ) ones. In those cases where one of the coefficients is much smaller than the other two, the two independent variables fit $Y/G=c_{R}R/G+c_{B}B/G+c_{G}$ can be satisfactorily approached by one variable fits as, e.g., $Y/G=c_{R}R/G+c_{G}$ or $Y/G=c_{B}B/G+c_{G}$, depending on the case, an approach that was basically followed in [51] although using different variable normalizations (i.e., G/R and B/G, respectively). The two independent variables fitting procedure presented in this work allows, however, for reducing the fit residuals at no relevant additional computational cost.

At first glance, the chloropic/G vs. X fit in the right-hand column of Figure 3 could seem to be slightly worse than the remaining ones. Note, however, that the vertical axis of this figure spans a range of values noticeably smaller than those of the other plots. Indeed, this fit is the one with smallest residuals (see Table 1 for Ch). Besides, the $c_{R}$ and $c_{B}$ coefficients are noticeably smaller than $c_{G}$, which in turn is of order 1. The reason is that the chloropic and G spectral sensitivity curves are very similar, so that the chloropic exposure turns out to be mostly determined by G, with some minor R and B adjustments to account for the different spectral content of the different types of lamps.

With regard to the negative sign of several of the fitting coefficients in Table 1, note that we are working in a (R/G, B/G, Y/G) space, each point of which corresponds to a given lamp spectral signature, irrespective of its absolute radiant flux. The set of lamps under consideration also does not completely fill the whole 3D space. Rather, the lamp points are contained within a bounded subspace, which turns out to be satisfactorily approximated by a 2D surface (in our case, and as a first fitting option, a plane). Moving from any point of this subspace to a neighboring one with higher value of R/G means considering a new lamp whose spectrum has comparatively more power within the R band (relative to the G one) than the original spectrum. Whether a given alfa-opic Y/G ratio will increase or decrease as a consequence of this change (or, equivalently, whether its $c_{R}$ fitting coefficient will be positive or negative) will depend on whether the new spectrum has comparatively more or less power in the alfa-opic Y band (relative to G) than had the original one. For typical lamp spectra it would not be unexpected that an increase of R/G could give rise to an effective decrease of the Y/G ratio for some alfa-opic channels centered in the blue region of the spectrum. This is consistent with the negative values of the $c_{R}$ coefficients for the Cy, Me, and Rh channels found in Table 1. The same effect (but swapping the spectral regions) could be expected for the $c_{B}$ coefficient of the Er channel, as is also observed in this Table. Therefore, negative fitting coefficients simply reflect the fact that, for the set of common lamp technologies used in this study, the cloud of points in the (R/G, B/G, Y/G) space is well fitted by a plane tilted toward the spectral region where the alfa-opic channel is centered. Negative values of the Y/G exposures are avoided because these fitting coefficients are only valid for the range of (R/G, B/G) defined by the sample of lamps. Any extrapolation of the Y/G fits outside this (R/G, B/G) subdomain is in principle not allowed.

Note that although we have chosen the spectral radiance L(λ) as the most appropriate input function due to its basic role in radiometry, the formal developments described in this paper can equally be applied, as previously mentioned, to any radiant magnitude of interest (e.g., the spectral irradiance, the radiant flux, or the radiant exposure) with no change in the expressions nor in the algebraic procedures. It has become a customary practice in experimental studies of non-image-forming effects of light to report as input variable the spectral irradiance at the corneal vertex plane, instead of the most basic spectral radiance. This corneal spectral irradiance is taken as a proxy for the spectral irradiance incident on the photoreceptors (after pre-receptoral filtering has been conveniently accounted for). As a matter of fact, however, the actual validity of this proxy depends on choosing very specific experimental illumination conditions. In the general case it is the corneal radiance and not the corneal irradiance that determines the retinal irradiances and, consequently, the actual photoreceptor exposures.

The present study has been made for the alfa-opic spectral sensitivity functions defined in [16,17]. CIE has recently issued a new standard (S 026/E:2018) for the metrology of optical radiation for ipRGC-influenced responses to light [54], where slightly different cone and rod sensitivity functions are proposed (their alfa-opic curves are replaced by the very similar S-, L-, and M- cone fundamentals, and the scotopic sensitivity curve), whereas the ipRGC (melanopic) function remains unchanged. The method described here can be adapted with no difficulty to this similar set of spectral curves.

Three main restrictions limit the scope of this work. First, only first order linear regressions have been used to determine the optimum Y versus R, G, and B estimators. Other approaches involving higher powers of R, G, and B could additionally be considered, with the proviso that over-fitting has to be avoided. Second, our study has been restricted to the spectra of a large but particular series of lamps widely used in lighting applications, both indoor and outdoor. Note that the R, G, and B to alfa-opic conversion coefficients obtained by linear fits are dependent on the statistical composition of the lamp sample, hence the values provided in Table 1 are applicable to the particular mixture used in our example (mostly composed of LED with low presence of HPS sources), and its validity for other lamp compositions should not be taken for granted. It is advisable that researchers using this approach recalculate the $\left(c_{R},c_{G},c_{B}\right)$ according to the a priori knowledge available about the relative percentage of each lamp type in their actual samples. Third, actual human light exposure does not depend only on the lamp spectra, but also on the spectral reflectance of the environment surrounding the observer (walls, façades, etc.). The key role of the environment in shaping the resulting spectral irradiance at the entrance pupil of the observer’s eyes has been theoretically and experimentally verified [55]. Wide-field DSLR panoramic images have the potential to provide estimates of the overall corneal band-weighted irradiance by means of the solid-angle integration of the radiance contributions of each pixel located in the hemisphere in front of the observer. In this way, we can account for the fact that not only the direct light from the sources but also the light reflected from the walls and other material media around the observer contributes to building up the total light exposure. Evaluating the DSLR performance for this application remains the subject of future work.

Notwithstanding these limitations, our results suggest that RGB photometry with DSLR cameras can be a useful technique for obtaining relevant information on the physiological inputs to the eye in the lit nightscape. Although DSLR cameras cannot compete in spectral resolution with hyperspectral devices such as the ones that have been successfully employed in this field of research (see [56] regarding the use of hyperspectral cameras for urban lamp identification in New York and [57] for their application to the determination of the streetlights band-weighted radiances, Cy, Me, Rh, Ch, and Er in the Barcelona nightscape and the resulting band-weighted irradiances at the eyes of the city dwellers), DSLR cameras offer clear advantages in terms of overall sensitivity and cost, with the same high angular resolution capability. The fact that a high number of photographers, astrophotographers, and citizen scientists alike, besides the professional light pollution research community, are frequent DSLR users can help to significantly expand the present capacity of acquiring extensive datasets for assessing physiological responses, which is information of key importance for epidemiologic studies addressing the health effects of human exposure to light at night. 

## Figures and Tables

**Figure 1 jimaging-05-00049-f001:**
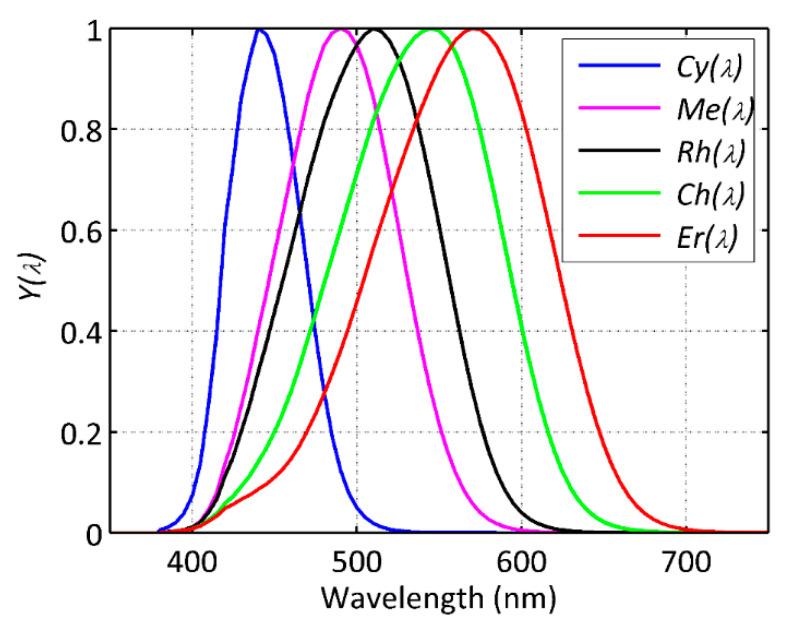
Human photoreceptor spectral sensitivity functions Cy(λ), Me(λ), Rh(λ), Ch(λ), and Er(λ), with SI compliant normalization [17].

**Figure 2 jimaging-05-00049-f002:**
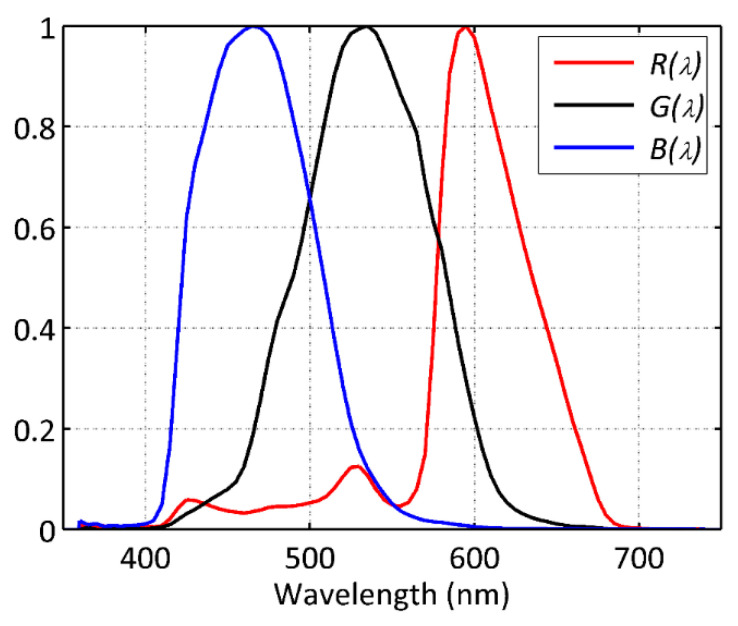
R(λ), G(λ), and B(λ) functions of a Nikon D3s camera, measured at LICA facility (Laboratorio de Instrumentación Científica Avanzada of Universidad Complutense de Madrid).

**Figure 3 jimaging-05-00049-f003:**
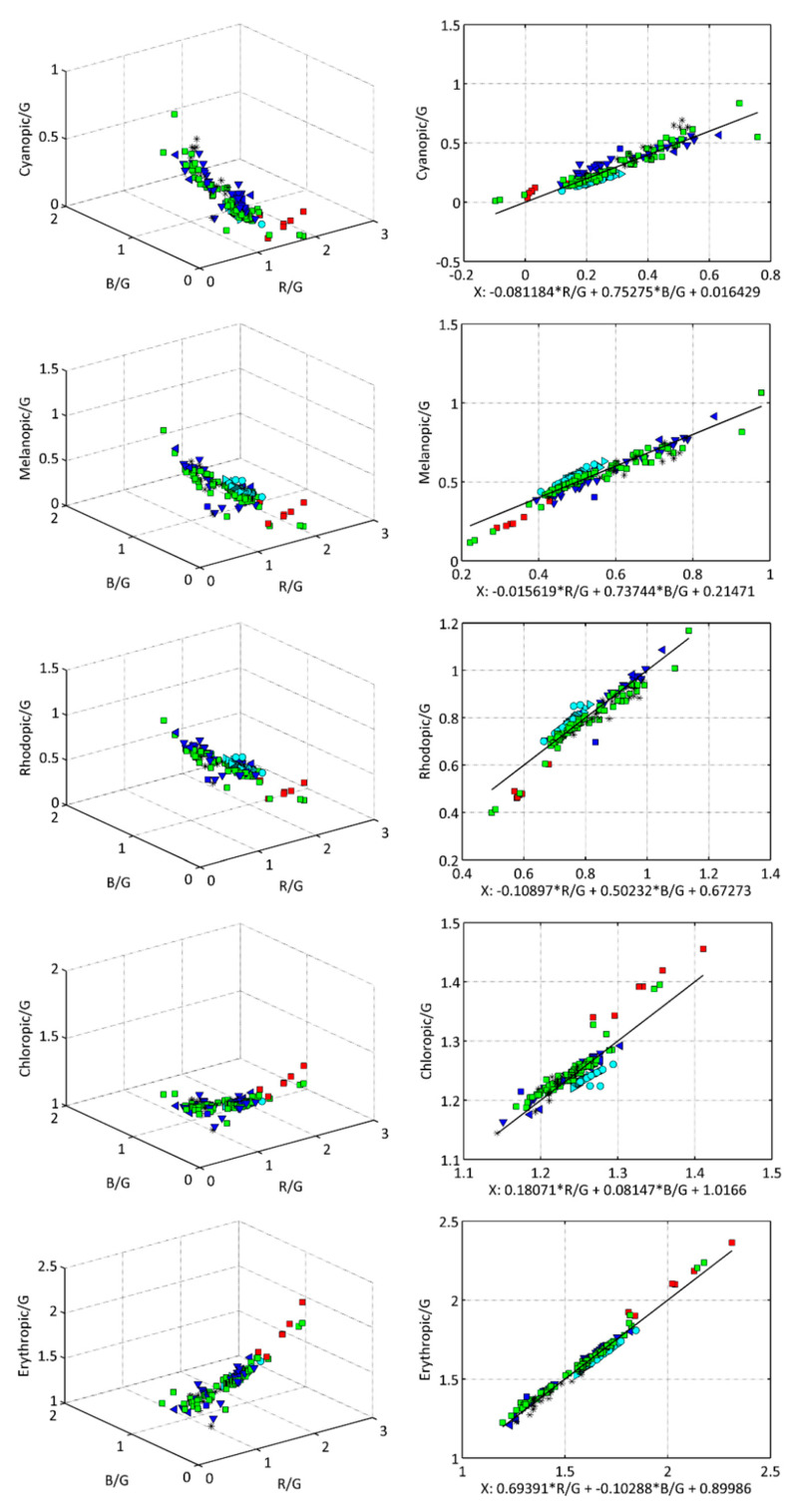
The G-normalized cyanopic (Cy/G), melanopic (Me/G), rhodopic (Rh/G), chloropic (Ch/G), and erythropic (Er/G) weigthed radiances as: (**left column**) points in the (R/G, B/G, Y/G) space; (**right column**) scatter plots of Y/G versus the optimum linear combination $X=c_{R}R/G+c_{B}B/G+c_{G}$. Lamp types: HAL(▶), INC(●), HPS(

), LED(

), CFL and FL(▼), CMH(◀), MH(✱), MV(

).

**Table 1 jimaging-05-00049-t001:** Optimum fitting coefficients ^1^ of $Y=c_{R}R+c_{G}G+c_{B}B$ for the five human photoreceptors.

Band/Coefficients	$c_{R}$	$c_{G}$	$c_{B}$	Residuals (std)
Cy	−0.0812(3)	0.0164(5)	0.7528(5)	0.0527
Me	−0.0156(3)	0.2147(5)	0.7374(5)	0.0409
Rh	−0.1090(5)	0.6727(7)	0.5023(7)	0.0358
Ch	0.1807(7)	1.0166(11)	0.0815(10)	0.0183
Ery	0.6939(8)	0.8999(13)	−0.1029(11)	0.0270

^1^ Note that these coefficients strictly apply to the particular lamp dataset used in this study. Samples with different lamp composition may require the use of different fitting coefficient values. The n-digit values in parentheses represent the standard deviation of the last n-digits of each fitting coefficient, due to the propagation of the uncertainties in the measurements of the lamp spectra, for a signal-to-noise ratio of 100:1.

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
