# Peer review of "Evaluating Human Photoreceptoral Inputs from Night-Time Lights Using RGB Imaging Photometry"

_2313-433X, 2019, doi:10.3390/jimaging5040049_

Reviewer 1 Report

The manuscript Evaluating human photoreceptoral inputs from night-time lights using RGB imaging photometry is an original, important and very well written contribution to the rising transdisciplinary field of study of the impacts of artificial light at night (ALAN). It describes a novel and accessible, though robust method to assess the effects of artificial light at night namely by the interaction of ALAN with the currently known visual and non-visual photoreceptors of the human eye.

The method was applied to a fairly large sample (205) of commonly used light sources. The manuscript also carefully addresses the corneal radiance as the determinant measurement, as opposed to the corneal irradiance, on the assessment of the actual exposure on the eye photoreceptors.

Moreover, the described method can also be straightforwardly adapted to other radiometric quantities, thus bringing added value to it.

In the discussion, the authors correctly address restrictions that, nevertheless, should not compromise the results at this level.

Apart from the two minor comments (please see below) and one question I would be happy to see answered by the authors, I have found no errors nor typos or actually nothing worth being modified. Great care was clearly given by the authors to the conception, writing and self-review of the manuscript. Overall, I was very pleased to review this work. 

Generic question: 

Fig. 3. Although the fit is good in all the graphs, do the authors have any explanation for the apparently slightly worse fit of the Chloropic/G vs. optimum linear combination X=cR/G + cB/G + cgraph?

Minor comments:

Lines 126-127: Given that the graph of Figure 2 is not a generic one but the result of the measurements for a particular camera model (a Nikon D3s - please see Figure 2’ caption), a reference to that camera model should probably be made in the main text, possibly on this same sentence. Alternatively, on line 127, the call to Figure 2 could be presented as an example of the R(λ), G(λ), and B(λ) functions of a camera (e.g. “(see example on Figure 2)”).

Lines 247-250: A shorter reference to the applicability of the method to other radiometric quantities has been made before (lines 123-124). The reinforcement here should possibly acknowledge that previous reference (e.g. “(…) as previously mentioned (…)” or similar).

Author Response

Comments and Suggestions for Authors

The manuscript Evaluating human photoreceptoral inputs from night-time lights using RGB imaging photometry is an original, important and very well written contribution to the rising transdisciplinary field of study of the impacts of artificial light at night (ALAN). It describes a novel and accessible, though robust method to assess the effects of artificial light at night namely by the interaction of ALAN with the currently known visual and non-visual photoreceptors of the human eye.

The method was applied to a fairly large sample (205) of commonly used light sources. The manuscript also carefully addresses the corneal radiance as the determinant measurement, as opposed to the corneal irradiance, on the assessment of the actual exposure on the eye photoreceptors.

Moreover, the described method can also be straightforwardly adapted to other radiometric quantities, thus bringing added value to it.

In the discussion, the authors correctly address restrictions that, nevertheless, should not compromise the results at this level.

Apart from the two minor comments (please see below) and one question I would be happy to see answered by the authors, I have found no errors nor typos or actually nothing worth being modified. Great care was clearly given by the authors to the conception, writing and self-review of the manuscript. Overall, I was very pleased to review this work.

Answer: We sincerely acknowledge this encouraging appraisal of our work. Please find below the answers to the generic question and minor comments.

Generic question:

Fig. 3. Although the fit is good in all the graphs, do the authors have any explanation for the apparently slightly worse fit of the Chloropic/G vs. optimum linear combination X=cR R/G + cB B/G + cG graph?

Answer: That is a very interesting question. The key word here is 'apparently'. Note that the vertical scale in the Chloropic/G vs optimum linear combination X spans a range of values significantly smaller than the other plots. As a matter of fact this fit is the one with the smallest residuals (see Table 1 for Ch). Note also that the cR and cB coefficients are noticeably smaller than cG, which in turn is of order 1. The reason is that the Chloropic and G spectral sensitivity curves are very similar, so that the Chloropic exposure turns out to be almost equal to the G one, with minor adjustments due to the different red and blue content of the different types of lamps. An explanatory note has been included in the revised version, lines 260-267 of the discussion.

Minor comments:

Lines 126-127: Given that the graph of Figure 2 is not a generic one but the result of the measurements for a particular camera model (a Nikon D3s - please see Figure 2’ caption), a reference to that camera model should probably be made in the main text, possibly on this same sentence. Alternatively, on line 127, the call to Figure 2 could be presented as an example of the R(?), G(?), and B(?) functions of a camera (e.g. “(see example on Figure 2)”).

Answer: Thank you for pointing out this issue. We have added that comment in line 128.

Lines 247-250: A shorter reference to the applicability of the method to other radiometric quantities has been made before (lines 123-124). The reinforcement here should possibly acknowledge that previous reference (e.g. “(…) as previously mentioned (…)” or similar).

Answer: We have included this reinforcement in line 292 of the revised version.

Reviewer 2 Report

This study considers how to convert R, G, and B readings from a DSLR camera to human photoreceptoral inputs in the S-cone, ipRGC, rod, M-cone, and L-cone. The authors find these relationships by performing linear regression fit to the synthetic photometric points derived from hundreds of lamp spectra. This is a great study that will enable the research community to use DSLR camera measurements to approximate the illuminance levels perceived by different photoreceptors in human eyes.  

Major comments:

1.      Did the authors characterize how much the best-fit lines would change among different types of lamp spectrum? If so, authors should include this information. If not, authors should mention this limitation in the Discussion that these conversion parameters are specifically for the mixture of lamps used in this study and might not be accurate to generalize it to all cases especially when the percentage of the types of lamp from a new study is very different from the lamp mixture used in this study.  

2.      In Table 1, several optimum fitting coefficients are negative. Physically, there should not be any negative coefficients. Authors need to constrain the coefficients to be greater or equal to zero while finding the best fit. Otherwise, authors need to explain why negative coefficients are accepted.

Minor comments

3.      For the lamp spectra used in this study, what is the relative percentage of different lamp types? Authors need to clarify this information because these best-fit lines are directly related to the relative percentage of lamp types considered in this study. For example, the best-fit lines can be very different if 90% of the lamps are LEDs vs. 90% are HPS lamps.

4.      A word of caution for the authors’ first main limitation stating that only first order linear regressions have been used: Higher-ordered fitting might not be suitable as it might over-fit the data. Thus, before considering using higher ordered functions and stating that the linear fit is a limitation, authors should investigate the goodness of fit for the low-ordered models first.

Author Response

This study considers how to convert R, G, and B readings from a DSLR camera to human photoreceptoral inputs in the S-cone, ipRGC, rod, M-cone, and L-cone. The authors find these relationships by performing linear regression fit to the synthetic photometric points derived from hundreds of lamp spectra. This is a great study that will enable the research community to use DSLR camera measurements to approximate the illuminance levels perceived by different photoreceptors in human eyes

Answer: We sincerely acknowledge this encouraging appraisal of our work. Please find below the answers to the major and minor comments.

Major comments:

1. Did the authors characterize how much the best-fit lines would change among different types of lamp spectrum? If so, authors should include this information. If not, authors should mention this limitation in the Discussion that these conversion parameters are specifically for the mixture of lamps used in this study and might not be accurate to generalize it to all cases especially when the percentage of the types of lamp from a new study is very different from the lamp mixture used in this study. 

Answer: This is a very relevant issue that was not explicitly addressed in the first version of this manuscript. We have included a specific comment in the discussion that reads:

"Second, our study has been restricted to the spectra of a large but particular series of lamps widely used in lighting applications, both indoor and outdoor. Note that the , ,  to alfa-opic conversion coefficients obtained by linear fits are dependent on the statistical composition of the lamp sample, hence the values provided in Table 1 are applicable to the particular mixture used in our example (mostly composed of LED with low presence of HPS sources) and its validity for other lamp compositions should not be taken for granted. It is advisable that researchers using this approach recalculate the  according to the a priori knowledge available about the relative percentage of each lamp type in their actual samples."

2. In Table 1, several optimum fitting coefficients are negative. Physically, there should not be any negative coefficients. Authors need to constrain the coefficients to be greater or equal to zero while finding the best fit. Otherwise, authors need to explain why negative coefficients are accepted.

Answer: Also a very interesting issue, worthy of some additional comments. We consider these negative coefficients should be accepted. Note that we are working in a (R/G, B/G, Y/G) space, each point of which corresponds to a given lamp spectral signature, irrespectively of its absolute radiant flux. Note also that the set of lamps under consideration does not completely fill the whole 3D space. Rather, the lamp points are contained within a bounded subspace, which turns out to be satisfactorily approximated by a 2D surface (in our case, and as a first fitting option, a plane). Moving from any point of this subspace to a neighboring one with higher value of R/G means considering a new lamp whose spectrum has comparatively more power within the R band (relative to the G one) than had the original spectrum. Whether a given alfa-opic Y/G ratio will increase or decrease as a consequence of this change (or, equivalently, whether its  fitting coefficient will be positive or negative) will depend on whether the new spectrum has comparatively more or less power in the alfa-opic Y band (relative to G) than had the original one. For typical lamp spectra it would not be unexpected that an increase of R/G could give rise to an effective decrease of the Y/G ratio for some alfa-opic channels centered in the blue region of the spectrum. This is consistent with the negative values of the  coefficients for the Cy, Me and Rh channels found in Table 1. The same effect (but swapping the spectral regions) could be expected for the  coefficient of the Er channel, as is also observed in this Table. So, negative fitting coefficients simply reflect the fact that, for the set of common lamp technologies used in this study, the cloud of points in the (R/G, B/G, Y/G) space is well fitted by a plane tilted toward the spectral region where the alfa-opic channel is centered. Negative values of the Y/G exposures are avoided because these fitting coefficients are only valid for the range of (R/G, B/G) defined by the sample of lamps. Any extrapolation of the Y/G fits outside this (R/G, B/G) subdomain is in principle not allowed.

These comments have been included in the discussion section of the revised version of the manuscript (lines 268-289).

Minor comments

3. For the lamp spectra used in this study, what is the relative percentage of different lamp types? Authors need to clarify this information because these best-fit lines are directly related to the relative percentage of lamp types considered in this study. For example, the best-fit lines can be very different if 90% of the lamps are LEDs vs. 90% are HPS lamps.

Answer: Yes, the reviewer is absolutely right. We have made explicit the number of lamps of each type in subsection 2.3 of the revised version. A note of caution has been included in the footer of Table 1. See also lines 313-319 of the enlarged discussion.

4. A word of caution for the authors’ first main limitation stating that only first order linear regressions have been used: Higher-ordered fitting might not be suitable as it might over-fit the data. Thus, before considering using higher ordered functions and stating that the linear fit is a limitation, authors should investigate the goodness of fit for the low-ordered models first.

Answer: We also agree with this warning. A note has been added in line 312 of the revised version.

Reviewer 3 Report

Digital camera measurements provide a novel tool for the characterisation of the effects of artificial light at night. The paper presents an important step for the interpretation of the measurements, the transformation from RGB colours to some physiologically important sensitivity functions.

The paper is straightforwardd, easy to read. I have only one criticism, and ask a minor correction related to this issue: It is essential to provide the possible errors and the level of systematic and statistical errors. The authors give the coefficients of the transformation matrix for 4-5 digit precision. It is clear that real life measurements do not have this level of accuracy, and also, these coefficients depend on the selected learning set (the used spectra). Please provide an error estimate for the coefficients and provide the numbers only for the significant digits.

Some minor points: A generally used method to calculate the Moore-Penrose inverse is the Singular Value Decomposition (SVD). This method is readily available in lots of mathematical software. Perhaps it is a good idea to mention it.

A recent CIE standard (S 026, 2018) also provide the ipRGC action spectrum. It may be useful to mention this standard and note the compatibility with the used action spectrum.

Author Response

Comments and Suggestions for Authors

Digital camera measurements provide a novel tool for the characterisation of the effects of artificial light at night. The paper presents an important step for the interpretation of the measurements, the transformation from RGB colours to some physiologically important sensitivity functions. The paper is straightforwardd, easy to read.

Answer: We also acknowledge this encouraging appraisal of our work. Please find below the answers to the criticism regarding uncertainties and the minor comments.

 I have only one criticism, and ask a minor correction related to this issue: It is essential to provide the possible errors and the level of systematic and statistical errors. The authors give the coefficients of the transformation matrix for 4-5 digit precision. It is clear that real life measurements do not have this level of accuracy, and also, these coefficients depend on the selected learning set (the used spectra). Please provide an error estimate for the coefficients and provide the numbers only for the significant digits.

Answer: The uncertainties in the estimated coefficients have been included in Table 1 of the revised version, with an explanatory footer note.

Some minor points: A generally used method to calculate the Moore-Penrose inverse is the Singular Value Decomposition (SVD). This method is readily available in lots of mathematical software. Perhaps it is a good idea to mention it.

Answer: A short note calling attention to this fact has been included in lines 186-188.

A recent CIE standard (S 026, 2018) also provide the ipRGC action spectrum. It may be useful to mention this standard and note the compatibility with the used action spectrum.

Answer:  Yes, this is a very interesting observation. As a matter of fact, the ipRGC spectrum used in our work and the one in CIE S026:2018 are identical. The cone and rod sensitivity functions however, are slightly different, having being substituted in this recent standard by the well-known CIE s-, l-, and m-cone fundamentals and the scotopic spectral sensitivity function. The differences between both sets of curves, however, are relatively small. A note has been included in lines 302-308 of the discussion section.

Round  2

Reviewer 1 Report

All issues were clearly explained or corrected.

Regarding the fit of the Ch/G vs. X (graph), in fact the residues are the lowest of all and the scale is not equal in all the graphs. Nonetheless, although in that sense not strictly necessary, I believe the explanation provided by the authors might now help to a faster interpretation of those graphs.

Reviewer 2 Report

The revised manuscript has addressed all my comments. It looks great, and I recommend "Accept in present form."

J. Imaging EISSN 2313-433X Published by MDPI AG, Basel, Switzerland RSS E-Mail Table of Contents Alert
Back to Top